# Clinical Pharmacy Services Enhanced by Electronic Health Record (EHR) Access: An Innovation Narrative

**DOI:** 10.3390/pharmacy10060170

**Published:** 2022-12-05

**Authors:** Zach J. Krauss, Martha Abraham, Justin Coby

**Affiliations:** 1Pharmacy Practice Department, School of Pharmacy, Cedarville University, Main Campus, Cedarville, OH 45314, USA; 2Cedar Care Village Pharmacy, Cedarville, OH 45314, USA

**Keywords:** community pharmacy services, pharmacists, public health, medication therapy management, electronic health record

## Abstract

Background: Patient care in the community pharmacy setting is often hindered due to limited access to adequate patient health information (PHI). Various studies suggest that lack of access to PHI is a main reason for delay in pharmaceutical care, medication dispensing errors, and lacking interprofessional relationships between prescribers and pharmacists. Literature has shown that interprofessional collaboration and improved access to PHI can improve transitions of care and communication for pharmacists, but literature is sparse on implementation of electronic health record (HER) access within independent community pharmacies. Methods: This observational study follows implementation of HER access into a rural community pharmacy to enhance common clinical services carried out by pharmacy staff. Metrics include number of enhanced consultations by pharmacy staff, type of consultations provided, potential reimbursement, decreased need to follow up with other providers, potential for decreased time to treatment or refills, and aspects of EHR most utilized during search. Results: Two-hundred sixty three patients’ profiles were assessed, with 164 (62.4%) deemed appropriate for EHR access and searching. Most interventions made were related to cardiovascular, endocrinologic, neuropsychiatric, and COVID-19 therapy medications. Conclusion: EHR access in community pharmacy has the potential to improve both the quality and availability of clinical patient interventions through enhanced knowledge of PHI.

## 1. Introduction

Increasing pharmacists’ access to patient health information has been proven to improve patient outcomes through pharmacist-led interventions [1,2,3]. Specifically, the addition of medication therapy management (MTM) review services has allowed pharmacists the opportunity to complete a comprehensive drug utilization review of all medications prescribed for patients, by providing a complete list of all medications the patient has filled in any pharmacy. Prior to these services, the pharmacist only had access to the list of medication that had been dispensed at their pharmacy alone, which could lead to issues of medication overutilization and/or non-adherence [1]. Since the integration of MTM and like-services, it has been shown that pharmacist-led, patient-centered interventions have a direct impact on patient medication adherence and overall disease control [2]. Disease states that showed direct benefit from pharmacist-led interventions included hypertension and cholesterol management, chronic obstructive pulmonary disease and asthma control, as well as overall medication knowledge and competency [2]. Literature has also shown that community pharmacist interventions were associated with enhanced clinical outcomes for type-2 diabetic patients (improved HbA1c measurements, symptomatology) in addition to their adherence to their antidiabetic regimens [3].

The logical next step to improving community pharmacists’ access to PHI is through the addition of electronic health record (EHR) access in the community pharmacy setting. Addition of EHR access would allow for a complete picture of the care each patient receives in both inpatient and outpatient medical settings, providing a clear view of the patient’s recent vitals, labs, and diagnoses that support the prescribing of a specific medication. Additionally, EHR access potentiates the community pharmacist in assisting the patient’s primary care provider in appropriate escalation or de-escalation of medications for chronic conditions.

Literature has shown that pharmacists in the community setting believe that lack of access to patient health information in the form of an EHR is a large barrier to providing quality clinical interventions [4,5,6]. Partially responsible for this barrier are ineffective reporting by MTM programs for issues that have already been addressed by a patient’s primary care provider, as well as difficult communication when trying to clarify patient issues when verifying a given patient’s prescription [7]. Additionally, often when a patient has been discharged from the hospital, the community pharmacies that dispense the patient’s medications are not provided a report of the updated medication list, leaving a gap in knowledge of what medications to initiate, continue, or discontinue upon discharge [6,8]. This gap in knowledge has been found to impact readmission rates and financial burden for patients [9,10].

Similarly, when observing trends in pharmacists’ preferences for communication platforms in settings such as transitions of care discussions and patient interventions through the community settings, pharmacists preferred using an EHR as opposed to other less structured forms of communication, such as faxing or using telephone calls [11,12]. Research has shown that medication errors and related prescribing issues are often a result of decreased communication, reduced access to patient-centered technology, and hindered collaboration between providers [13]. Health information exchange (HIE) generally refers to various methods used in healthcare to expand secure access to patient medical information with the intent to improve patient care [14]. HIE has been used in some pharmacy settings in order to receive part of the patient record in order to assist with services such as transitions of care [15]. Similar strategies of utilizing HIE to expand EHR access have been published to inform practitioners on innovative practice and how expansion might occur on a local level [16,17]. In such literature, access to HIE and EHR information has been proven to allow pharmacists to enhance preventive medication services and to recognize medication-related problems [17].

There is a sparsity in the literature regarding the impact of EHR access for community pharmacists, however, and thus it is difficult to truly assess the utility and cost-effectiveness of system-wide integration of EHRs into community pharmacies. While many articles exist discussing the community pharmacists’ desire to have access, there is limited knowledge of how much EHR access impacts patient care in the outpatient pharmacy setting. This narrative commentary aims to describe the impact of integrated EHR access in a rural community pharmacy setting and the effects that access had on the pharmacists’ ability to care for patients well when problems have arisen such as during medication therapy management (MTM) conversations or at prescription pickup. It also showcases EHR access benefits in addressing issues before they become problematic in the form of a hospitalization or error in transition of care.

The overall objective of this article is to showcase the utility of EHR access in a small community pharmacy and to describe the potential impact widespread access to EHR may have on the health of patients within the pharmacy.

## 2. Materials and Methods

Access to patients’ electronic health record profile was provided through a system known as Parawell, which is sourced by Health In Motion Network, a medical, pharmacy-focused organization whose aim is to help community pharmacists create a better model internally for patient care management. In addition to providing a platform for pharmacists to engage with patients and create notes based on clinical interventions, the novel aspect of the platform is that it additionally provides access to real-time health data and records so that the pharmacist can see patient information without having to reach out to providers for clinical information.

Some of the information pharmacists have access to through Parawell includes historical lab values and vitals, patient chart notes and clinical assessments, list of medications, diagnoses, allergies, immunizations, in addition to other information you would typically discover in a standard EHR system. The patient information is collected by Parawell using claims information which is submitted to medical insurance. Patients without medical insurance or those partnered with insurance Parawell has not contracted with would not have information accessible in this case.

Implementation of Parawell into pharmacy clinical services workflow began on 6 June 2022, and clinical assessment was carried out in the pharmacy from 15 June 2022 through 15 September 2022 in order to assess efficacy, utility, and pharmacist perceptions on workflow impact over the course of the four-month period. Study was conducted in a single community pharmacy in the Midwest region of the United States of America, as part of a pilot project with the Health In Motion Network.

Clinical interventions were tracked by all pharmacists and pharmacy interns participating in clinical services. In order to assess patient perceptions on enhanced services, anecdotal narratives based on patient impact stories were also collected. All patient information was de-identified after collection and IRB exempt status was obtained through Cedarville University prior to data collection, indicating that appropriate steps were taken to protect the rights and welfare of participants.

No formalized screening process was utilized, rather, the majority of interventions were carried out on patients that were highlighted by the pharmacy dispensing system as either late to refill a medication, or someone who had a history of picking up medications late over a 6 month period. The dispensing system highlighted patients which the clinical staff then evaluated for potential intervention. If a patient was deemed appropriate based on history of fill and lack of other potential factors (such as students who may be filling with another pharmacy or individuals with a known hospitalization, etc.), they were deemed appropriate for screening through the Parawell system and EHR information was accessed. There were no explicit exclusion criteria, and only those who were active patients within the pharmacy were included for analysis.

During the course of implementation, each patient that was searched in the Parawell platform was tracked using an excel spreadsheet in order to determine the actual utility of access. After intervention tracking data collection, interventions were broken down into the following four categories: chronic disease state management, immunization administration, drug therapy management, and laboratory/vital assessment recommendations. While patient-focused clinical interventions were already being made within the pharmacy, the interventions were often limited in scope due to the low quantity of health information provided through prescriptions and patient recall. MTM services were carried out regularly before integration of EHR access, but often requests were only informed by fill history rather than full patient chart information from their system’s EHR system. Only interventions that were enabled or enhanced by EHR access were included in the results in order to showcase the utility of EHR access within the pharmacy.

## 3. Results

### 3.1. Overall Patient Encounters

From 15 June 2022 to 29 September 2022, 263 total patients were screened within the Parawell for potential outreach by pharmacy clinical staff. Of those, 164 (62.4%) unique patients were deemed appropriate for some kind of intervention. Of these patients, the vast majority of interventions made were related to MTM and drug utilization in the patient based on medication history (157, 95.7%). 5 patients’ (3.0%) information were pulled solely for immunization administration assessment, and 2 patients’ information (1.2%) were pulled specifically for chronic disease state management. All of the encounters for MTM and drug utilization were necessary and completed to assist in clinical decision making, involving whether a patient should or should not be dispensed a specific medication. Of the 164 patients in whom intervention was deemed necessary, 105 (64%) of those interventions involved reaching out to the patient either in person or by phone in order to discuss the issue encountered within the health record.

### 3.2. Immunization Administration

While only five encounters were solely initiated due to immunization administration history, a total of 48 patients were asked about immunization history in some capacity. Most of these discussions were initiated in addition to medication adherence calls, after reviewing their history and recognizing that they were missing a recommended immunization on ImpactSIIS. ImpactSIIS is the state immunization registry through the Ohio Department of Health. A total of 104 vaccinations were recommended through these encounters, with the most commonly recommended vaccines including any portion of the COVID-19 vaccination series (32, 30.7%), herpes zoster (30, 28.8%), and tetanus, diphtheria, and pertussis (TDAP, 27, 26%). Anecdotally, many patients reported not knowing or not having been told that they were due for routine vaccinations such as herpes zoster. Many of these patients did receive the recommended vaccine from our pharmacy staff, following the encounter.

### 3.3. Drug Therapy Management

As previously mentioned, a majority of interventions made through this program were related to medication therapy. Of the 157 interventions made in this category, patients were reached about a wide range of medication classes and questions, ranging from confirming doses and adherence in patients with limited medication history, to confirming eGFR and kidney function for patients to be initiated on COVID-19 antiviral therapy. Groups of interventions were grouped by the following medication classes: anticoagulation, COVID-19/infectious disease, cardiology, endocrinology, psychiatric/neurologic, respiratory, and general, which included comprehensive medication reviews. Table 1 showcases all drug therapy management interventions stratified by the classification of medication addressed.

Within the drug therapy management interventions, one of the most helpful aspects of access to the EHR brought to the pharmacy staff was the ability to monitor patient histories for potential interactions or contraindications to COVID-19 antiviral therapy. The community pharmacy has been highly involved with test and treat protocols, specifically for the combination nirmatrelvir/ritonavir product in patients who are symptomatic and have a positive COVID test result through in-house point-of-care testing. Through searching patient charts, estimated glomerular filtration rate (eGFR), creatinine clearance (CrCl), and kidney function could be assessed without additional calls to the patient or their provider, allowing for safer and quicker prescribing. Additionally, the pharmacy staff could use hospital medication history information and historical fill data from the health system in order to cross-reference for potential drug–drug interactions with oral antiviral medication for treatment of COVID.

EHR access was found to be necessary specifically for interventions pertaining to cardiology, endocrinology, and anticoagulation medications, as pharmacy staff were able to access laboratory values and vitals that would otherwise not be available. Access to recent clinical measurements, like A1c or specific blood glucose measurements at certain times of the day assisted in determining appropriate recommendations for optimizing adherence and overall blood glucose control. Similarly, having access to INR/PT (international normalized ratio or prothrombin time) is indicative of a patient’s adherence or therapeutic sensitivity to anticoagulants, and having access to most recent blood pressure measurements assists in determining whether further optimization is necessary for antihypertensive regimens. This patient information becoming accessible to the pharmacy staff allowed for substantial improvement in the insight and preparation that could go into a conversation with a non-adherent patient.

Neuropsychiatric medications including attention-deficit, hyperactivity disorder (ADHD) medications, antidepressants, anti-seizure medications, and medications for Parkinson’s were commonly addressed during interventions. Access to patient EHR records allowed pharmacy staff to check to ensure doses were supposed to be adjusted rather than relying on patient reported information or history. Access also allowed staff to assess ICD-10 diagnosis codes in order to determine the specific indication for the prescribed medication, which allowed for more personalization of patient interaction.

Finally, medications such as inhalers, leukotriene receptor antagonists, and other respiratory agents were able to be assessed in patient charts in order to assess appropriate inhaler regimens, to assess what stage patients were in for their asthma or chronic obstructive pulmonary disease (COPD) diagnosis, and to also check history of hospitalization or exacerbation.

## 4. Discussion

The results of this study showcased over 160 interventions that were made possible or enhanced by EHR access over the period of the study timeline. For this rural pharmacy practice, this is greater than 15% of the total patient population served by the pharmacy. Interventions were mostly related to drug therapy management, specifically within the cardiology, endocrinology, and neurology/psychiatry therapeutic areas. Access to information related to previous visits, laboratory values, and diagnoses from hospital encounters helped clinical pharmacy staff to make interventions that otherwise would have been impossible without EHR access or would have required intensive communication with the patients’ care teams to clarify information.

### 4.1. Cost-Efficacy

This pilot did not attempt to integrate billing for services due to current scope of practice changes in the state of Ohio, changes in Ohio provider status provisions, and inability to integrate Parawell services and note documentation with billing services during the time of data collection. All patients that were reached out to, however, were assessed for potential billing value based on available Current Procedural Terminology (CPT) codes that are being implemented into government funded plans in Ohio. All codes utilized in this study were noted as outpatient visits for evaluation and management (E&M) of which three CPT codes were used based on timing of the call. 99211, 99212, and 99213 were used to estimate potential reimbursement for 5, 10, and 15 min phone interactions with patients. Table 2 outlines potential compensation based on the number of patient calls made and the timing of each call.

The world of pharmacy continues to expand, with the scope of practice of many pharmacists in settings such as clinical inpatient and primary care expanding to meet the needs of the ever-growing medical system. Pharmacists in community pharmacy have the potential of growing along with other sectors of pharmacies by pushing for access to PHI and implementing holistic care at the checkout counter.

Community pharmacists have expressed positive perceptions of EHR access especially when considering the impact it can have on patient-centered care [18]. Excessive need for phone or fax communications with providers after prescriptions are sent to pharmacies can create both frustration and risk in the pharmacy setting when clarifications on prescriptions are necessary [19]. Being granted access to EHR platforms for personal use by clinical pharmacists has shown improved adherence to guideline-directed care and overall improvement in communication between pharmacy and medical staff [20]. Innovations such as EHR platforms, HIE, immunization history systems, etc., have showcased benefits for community pharmacy practice and have the potential to improve patient safety [21]. Pharmacists have been shown to be able to provide better, more confident assessment of patient and their drug therapy problems when EHR information is accessible [22].

The information collected by clinical staff allowed for more integrated assessment of patient needs and for more intentional tailoring of patient care. Snyder et al. reported that community pharmacist collaboration with primary care offices in order to gain access to electronic health records helped to improve adherence to guideline-directed care and the access provided was integral for quality pharmacist recommendations and communication [20]. A report of utilization of My Health Record, an EHR system for patients and healthcare professionals rolled out in Australia, showcased strong utility for the system for improving both safety and quality of patient interventions [23]. A similar report showcased the benefit of My Health Record for also preventing relapse and hospitalization by setting up clinical decision support [24]. Our report showcases patient information that otherwise would not have been accessible for pharmacists and clinical staff through the combination of electronic health record data with local pharmacy prescribing and fill data. From qualitative research done in Ohio on pharmacists’ perceptions of integration, common barriers identified included lack of time to participate, inadequate access or opportunity to review EHR information, lack of or inadequate reimbursement, and limited support staff availability [25].

One important aspect to research when assessing the potential of EHR integration is cost of the service. Whether through HIE or complete integration of an EHR system, the pharmacy will most likely be paying for the EHR program which could potentially be a barrier to expansion. Significant cost becomes apparent when new information technology (IT) schemes must be introduced into a system, especially when the system is independent and not relying on a large health-system [26].

Integration of EHR access into a community pharmacy has other potential barriers that have been previously outlined in the literature, as well. Some of these barriers include lack of standardization, decreased productivity and workflow disruptions, and difficulty communicating between EHR software and in-house pharmacy software [27]. Another potential barrier to integration of EHR would be provider and medical staff perception of HIE within the pharmacy sector. Issues such as lack of knowledge, concerns for implementation issues, and regulatory concerns have emerged as potential barriers to full provider support of HIE, especially fully automated HIE within the pharmacy sector [28]. Barriers for pharmacists are also a reasonable concern with integration of new EHR access through HIE.

Another consideration for integration of EHR access into the pharmacy would be training of pharmacy staff on the clinical interventions and utilization of the electronic platform. The system being used in this study through Parawell is fairly user-friendly, but does take some baseline training in order to understand. Additionally, the process for finding and making clinical interventions without the assistance of an MTM platform can take some adjustment by the pharmacy staff. Research has shown implementation can be difficult when managing full-time-equivalent staff when interventions can on average take approximately 21 min each [29]. Finally, legal considerations and regulations must be taken into account as well, as each state has unique requirements for HIE, as well as for reimbursement for services provided by pharmacists. Many states have still not addressed HIE in legislation, which in itself can be a barrier to implementation [30]. Each pharmacy team would need to assess this before actualization of clinical programs utilizing HIE and EHR platforms.

This article has showcased the utility of integrated EHR access in a small community pharmacy with less than 1000 total patients in a rural community. Utilization of EHR by clinical pharmacy staff has assisted with better and more efficient answering of pharmacy-related questions, and allows for more clinical discussions with patients about their medication history. Research is necessary for other sizes of pharmacies catering to different demographics in order to assess the benefit these platforms are able to provide to patients in such settings. However, anecdotally, it is clear that there are opportunities for growth in community pharmacy clinical practice being created by access to these platforms.

### 4.2. Limitations

Potential limitations to this study is the limited patient population which is inherent to rural community pharmacies. The patient population of the pharmacy as a whole is approximately 1000 patients served each year, which limits the expansion of services and the number of total potential patients observed through the EHR platform.

One potential drawback of collecting patient information using the systems utilized in this specific setting is that patients who are uninsured or who have coverage from insurance companies that do not collaborate with Parawell will not have accessible information to the pharmacy team. This simply means that not all patients will be able to be accessed, so it is not always guaranteed that clinical services can be carried out through this modality.

## 5. Conclusions

In this study the research group was able to obtain EHR access as part of a pilot, at no cost to the pharmacy. EHR access in community pharmacy has the potential to improve both the quality and availability of clinical patient interventions through increased accessed to and knowledge of PHI. Maintenance of the program would likely be associated with cost, which is a definite consideration when determining long-term implementation of EHR programs in a small, rural, community pharmacy. In the spirit of continuous quality improvement, our pharmacy desires to maintain EHR access long-term to continue to build upon the determined impact through this study. So far, our pharmacy has not needed buy-in from the prescribers, but this is something necessary to consider if clinical services continue to expand. Future considerations on utilization of medical billing will be necessary for the pharmacy in order to assess cost-efficacy of the program. This could potentially counteract cost concerns mentioned in the discussion, and might allow for even more staffing within the pharmacy (in the form of clinical pharmacists or pharmacy interns) who could continue the trend of clinical patient interventions.

## Figures and Tables

**Table 1 pharmacy-10-00170-t001:** Drug Therapy Management Interventions by Category.

Category	N	%
Cardiology	55	35.0
Endocrinology	41	26.1
Neurology/Psychiatry	23	14.7
Infectious Disease/COVID-19	20	12.7
Anticoagulation	8	5.1
Respiratory	8	5.1
General	2	1.3

N = number of interventions in each category. % = percentage of interventions in each category

**Table 2 pharmacy-10-00170-t002:** Potential CPT Code Utilization, Reimbursement, and Overall Compensation for Services.

Numeric CPT Code	Evaluation and Management (E&M) Indication	Reimbursement	Carried Out	Potential Compensation
99211	Other outpatient visit for E&M of patient, 5 min	$12.32	18	$221.76
99212	Other outpatient visit for E&M of patient, 10 min	$22.72	55	$1249.6
99213	Other outpatient visit for E&M of patient, 15 min	$37.06	32	$1185.92
**Total**				**$2657.28**

## Data Availability

All data presented in this study are contained within this article. Supplemental data was not produced for this study but original data-sets with de-identified information may be requested from the corresponding author.

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
