# Peer review of "Clinical Pharmacy Services Enhanced by Electronic Health Record (EHR) Access: An Innovation Narrative"

_pharmacy, 2022, doi:10.3390/pharmacy10060170_

Round 1

Reviewer 1 Report

Dear authors,

I consider the current topic very pertinent, and an area in which there is still little evidence.

However, the methodology chapter is not enough clear and detailed.

There is no information regarding patients recruitment and selection. Also the source of health information used by the Parawell system should be clarified. Even more, the description is not enough in order to achieve the study purpose, and no details were included on how the impact of access to the EHR will be evaluated.

Considering the results presented, the established objectives for the current study were not answered (“describe the impact of integrated EHR access in a rural community pharmacy setting and the effects that access had on the pharmacists”).

The discussion of results is too sparse and should also be reviewed and improved.

Further details: in the Table 1 legend the word "Category" is duplicate; in the Table 2, for the acronyms used there is no meaning in the legend.

Reviewer 2 Report

Dear authors

Thank you for the opportunity to review this manuscript dealing with a highly relevant topic - the implementation of EHR access in community pharmacies. However, the manuscript need significant improvement and clarifications before publication. Below I summarize my main concerns that need extra attention in the revised version of the paper:

Introduction

- the background lacks a sufficient overview of relevant litterature. The manuscript only has 6 references in total and there are several sections and statements provided without any reference in the background.

- Please add something about the legal aspects of providing EHR data to community pharmacies

- some of the concepts you use should be explained more. For example what do you mean with medication therapy management (MTM)? Is it some kind of clinical decision support system. 

- several sentences are very long and difficult to follow - please check and revise for clarity

- be consistent with how you use abbreviations and explain it the first time. For example you mix writing EHR and electronic health record throughout the manuscript. In addition you write electronic medical record in the title.

Aim

- the objective/aim should be clarified. Right now you first write that the aim is to describe the impact of integrated EHR access and the effects on pharmacists ability to care for patients well. Later you write that the objective is to showcase the utility of EHR in community pharmacy and describe the potential impact it may have on public health. Which is it? Can you answer this with the data you have and the method you chose? 

- The aim/objective is missing from the abstract

- in the discussion you write about showcasing potential efficacy. In other parts of the paper you write about enhanced access.

Materials and methods

- clarify the study design and the methods. the title says "Innovation narrative". Is there a specific method and structure for conducting and reporting that? Please explain.

- I would like some more description about the setting. Which country/region? How large is this implementation of parawell? only this pharmacy? is it a pilot project?

- Are the clinical care servces new as well or is the only new thing the EHR access?

- please explain IRB approval. is it an ethical approval?

- in results you have a section about overall patient encounters where you describe a screening process. Please describe this in method. What were the inclusion and exclusion criteria

- Explain the anecdotal narratives. Where they collected from pharmacists or from patients? How? How did you analyze the anecdotal narratives? 

Results

- The first section of the results should be included in methods instead

- what do you mean with "enhanced patient"? first sentence of results.

- the results section is difficult to follow. Which encounters/interventions and outcomes were included? You write about 164 patient interventions and 5 of them being related to immunization, but then you include 48 patents in that analysis.

- are you only describing things that were enabled by the EHR access or all interventions

- page 4, In the first two sections it is not clear where the data comes from. for example the section starting with "another beneficial aspect of the EHR access...". Where does this result come from? clarify? is it an interpretation of the results it should be in the discussion. 

- the section about cost-efficacy is confusing. You start with writing that this was not within the scope but then you still provide lots of data that is difficult to interpret. If you want to keep this it should be included in the aim/research questions and explained in more detail. If not, remove it.

Discussion

- the discussion should begin with a short section summarizing the main findings of the study.

- discussion should be extended with more references

- Add a conclusion that answers the aim/objective you have and that is supported by your results. Now you only write that the study showcased efficacy of integrated EHR.

Best of luck! I'm looking forward to read more about this implementation and the results from it.

Reviewer 3 Report

In the results section authors gave the information that "164 unique patients were deemed appropriate for some kind of intervention" and next due to DTM only the total number of interventions (157) were given. It would be more interesting for readers to have also information for how many patients those DTM interventions were necessary. Also in table 1 such information in each category could be added. If one intervention = one patient authors should add clear statement and explanation about it.  

Round 2

Reviewer 1 Report

Dear authors,

Thank you for your effort in revising the manuscript.

Changes performed in the manuscript answered most of the questions raised by the reviewers. Despite the limitations in the study design, in order to draw conclusions about the pharmaceutical intervention using EMR access, I think that improvements were made in the various chapters, including the description of the methodology used, the presentation of the results and the discussion of them.

Reviewer 2 Report

Thank you for the revised version of the paper and response to the comments. The manuscript is significantly improved. I just have one thing that I think should be revised. The conclusion now almost entirely focus on cost which is not what you describe in the aim/objective. I suggest moving the section about cost to the discussion and write the conclusion from your study based on your objective in that section. I do not need to read the revised version again. For example the conclusion your write in abstract but with a little more information.
